# Utility of Liver Function Tests and Fatty Liver Index to Categorize Metabolic Phenotypes in a Mediterranean Population

**DOI:** 10.3390/ijerph17103518

**Published:** 2020-05-18

**Authors:** Dariusz Narankiewicz, Josefina Ruiz-Nava, Veronica Buonaiuto, María Isabel Ruiz-Moreno, María Dolores López-Carmona, Luis Miguel Pérez-Belmonte, Ricardo Gómez-Huelgas, María Rosa Bernal-López

**Affiliations:** 1Preventive Medicine Department, Virgen de la Victoria University Hospital, 29010 Malaga, Spain; darek_1980@yahoo.es; 2Internal Medicine Department, Regional University Hospital of Málaga, Instituto de Investigación Biomédica de Málaga (IBIMA), 29010 Malaga, Spain; jruiznava@hotmail.es (J.R.-N.); veronicabuonaiuto@yahoo.com.ar (V.B.); mruiz.salud@gmail.com (M.I.R.-M.); mdlcorreo@gmail.com (M.D.L.-C.); luismiguelpb@hotmail.com (L.M.P.-B.); 3Ciber Fisiopatología de la Obesidad y Nutrición. Instituto de Salud Carlos III, 28029 Madrid, Spain

**Keywords:** Non-alcoholic fatty liver disease, body mass index, lipids, diabetes

## Abstract

The aim of this study was to analyze the utility of liver function tests (LFT) and fatty liver index (FLI), a surrogate marker of non-alcoholic fatty liver disease, in the categorization of metabolic phenotypes in a Mediterranean population. A cross-sectional study was performed on a random representative sample of 2233 adults assigned to a health center in Málaga, Spain. The metabolic phenotypes were determined based on body mass index (BMI) categorization and the presence or absence of two or more cardiometabolic abnormalities (high blood pressure, low high-density lipoprotein (HDL) cholesterol, hypertriglyceridemia, pre-diabetes) or type 2 diabetes. No difference was observed between metabolically healthy and metabolically abnormal phenotypes on LFT. The mean FLI of the population was 41.1 ± 28.6. FLI was significantly higher (*p* < 0.001) in the metabolically abnormal phenotypes in all BMI categories. The proportion of individuals with pathological FLI (≥60) was significantly higher in the metabolically abnormal overweight and obese phenotypes (*p* < 0.001). On a multivariate model adjusted for sex, age, and waist circumference, a significant correlation was found between pathological FLI and metabolically abnormal phenotypes in the overweight and obese BMI categories. Area under the curve (AUC) of FLI as a biomarker was 0.76, 0.74, and 0.72 for the metabolically abnormal normal-weight, overweight, and obese groups, respectively. Liver biochemistry is poorly correlated with metabolic phenotypes. Conversely, a good correlation between FLI, as a marker of non-alcoholic fatty liver disease (NAFLD), and metabolically abnormal phenotypes in all BMI ranges was found. Our study suggests that FLI may be a useful marker for characterizing metabolically abnormal phenotypes in individuals who are overweight or obese.

## 1. Introduction

The liver is the largest gland in the human body and plays a role in many functions, such as bile production as well as absorbing and metabolizing bilirubin, fats, proteins, and carbohydrates. Due to the obesity pandemic, non-alcoholic fatty liver disease (NAFLD) is currently the most common cause of abnormal liver function tests (LFT) [1]. NAFLD affects approximately 25% of the global population [2] and up to 80% of people with obesity [3]. NAFLD is also associated with metabolic comorbidities such as type 2 diabetes (22%), hyperlipidemia (69%), hypertension (39%), and metabolic syndrome (MetS) (42%) [4]. Two described types of NAFLD, isolated hepatic steatosis and non-alcoholic steatohepatitis [5], affect 30% and 5% of the population, respectively [1] and can eventually lead to cirrhosis [6].

Among subjects with obesity, those with the highest amount of visceral fat [7] and those whose adipocytes are largest in diameter [8] are those at highest risk for NAFLD. The variation in the amount of visceral fat, however, can only explain 40% of the variation in liver fat content, indicating that there are other factors that regulate liver fat content [9]. The role of visceral fat in the development of NAFLD may be related to increased portal flow of fatty acids [10]; deregulation of adipokine secretion [11], with decreased hepatic fatty acid oxidation; and increased liver lipogenesis [12,13].

Various mathematical equations have been validated to estimate the grade of hepatic steatosis without having to resort to invasive tests (such as liver biopsy) or tests with limited accessibility (1H-magnetic resonance spectroscopy). Although they do not distinguish between the two forms of NAFLD, they are useful in screening for this disease. The calculation of the NAFLD score and liver fat percentage (LF%) is based on presence of MetS, type 2 diabetes, insulin, and aspartate aminotransferase (AST) level and AST/alanine transaminase (ALT) ratio [14,15].

Recent studies support the idea that liver enzymes can be useful markers for monitoring the progression of NAFLD [16]. NAFLD is characterized by an alteration in biochemical pattern, with increased levels of transaminases (ALT, AST, and gamma-glutamyltransferase (GGT)) [17]. However, it has been demonstrated that liver enzyme levels do not correlate with the histological severity of NAFLD [18]. Nevertheless, ALT, AST, the AST/ALT ratio, and GGT are included in various multi-biomarker panels aimed at optimizing the diagnostic accuracy of NAFLD [19].

The fatty liver index (FLI), described by Bedogni et al. [20], estimates the probability of NAFLD with an accuracy of 0.84 (95% confidence interval (CI): 0.81–0.87). It takes into account the following parameters: triglycerides, body mass index (BMI), GGT, and waist circumference. An FLI < 30 may be used to rule out NAFLD and an FLI ≥ 60 may indicate the disease.

The concept of metabolically discordant phenotypes is based on the fact that not all individuals with the same BMI have the same metabolic profile [21]. Thus, there are individuals who are metabolically healthy obese (MHO) and metabolically healthy overweight (MHOW), who do not develop the metabolic complications related to excess weight. On the other hand, there are individuals who are metabolically abnormal normal-weight (MANW), who are normal-weight but develop abnormalities such as hypertension, dyslipidemia, insulin resistance, and increased inflammatory markers. Nevertheless, MHO and MHOW are considered transient states; indeed, it has been found that over 33.3% of these individuals became metabolically abnormal obese after nearly 10 years of follow-up [21,22].

There is limited data in the medical literature about NAFLD in the different metabolic phenotypes. In a recent publication, excess adiposity, even without accompanying metabolic health status, may contribute to fibrosis progression in NAFLD [23]. Among all the methods to diagnose NAFLD, we emphasized FLI because it is a non-invasive test that is easy to perform and is highly accessible in clinical practice. On the other hand, not all individuals with the same BMI have the same metabolic profile and several studies have attempted to establish criteria characterizing each one. With this background, the aim of this study was to analyze the utility of liver biochemistry and FLI in the categorization of the metabolic phenotypes in a Mediterranean population.

## 2. Methods

To study the prevalence of MetS, we designed a cross-sectional epidemiological and analytical study in a representative random sample of the adult population (18–80 years) assigned to the Ciudad Jardin Health Center in Málaga (29,818 people). In order to calculate sample size, we assumed a MetS prevalence of 20% [24], a confidence level of 95% (alpha error 0.05), statistical power of 80%, and losses of 15%. After a six-month recruitment process, a sample of 2492 individuals was obtained. There were 154 people (6.2%) excluded for not meeting the inclusion criteria or for meeting the exclusion criteria: 37 (1.5%) had a BMI < 18.5 kg/m^2^, 56 (2.2%) could not be located, and 12 (0.5%) objected to participating in the study. The final sample included 2333 individuals (Figure 1).

The inclusion criteria were individuals aged 18–80 years assigned to the health center who were mobile and who provided written informed consent. Subjects were excluded if they had a severe associated disease; were terminally ill or immobile, pregnant, or hospitalized at the time of the study; or suffered from severe psychiatric disorders, alcohol use disorder, or drug addiction. Subjects were located by telephone and invited to participate in the study. After obtaining signed informed consent from those who wanted to participate, all the subjects underwent a clinical interview at their usual primary care office, where a physical examination was performed by trained healthcare workers (doctors or nurses) and a blood sample was drawn after a 12-hour fast. The samples were analyzed via the routine methods in the reference hospital laboratory.

We defined the metabolic phenotypes as follows. Based on BMI, we established four groups: underweight (BMI < 18.5 kg/m^2^), normal-weight (BMI 18.5–24.9 kg/m^2^), overweight (BMI 25.0–29.9 kg/m^2^), and obese (BMI ≥ 30.0 kg/m^2^). The metabolically abnormal phenotype was diagnosed when at least two of the following metabolic abnormalities were present: (1) systolic blood pressure ≥130 mmHg and/or diastolic blood pressure ≥85 mmHg (or use of antihypertensive drugs in an individual with a prior history of hypertension) [25]; (2) triglycerides ≥150 mg/dL (or treatment with fibrate, nicotinic acid, or omega-3 fatty acids) [25]; (3) high-density lipoprotein (HDL) cholesterol <40 mg/dL in men and <50 mg/dL in women (or treatment with a fibrate, nicotinic acid, or omega-3 fatty acids) [25]; (4) prediabetes defined as blood glucose levels 100–125 mg/dL and/or glycated hemoglobin A1c (HbA1c) 5.7%–6.4% [26]. Patients with type 2 diabetes were considered metabolically abnormal regardless of whether they had other metabolic abnormalities. Finally, based on BMI group and the presence or absence of a metabolically abnormal phenotype, we established six metabolic phenotypes: (1) metabolically healthy normal-weight (MHNW), (2) metabolically abnormal normal-weight (MANW), (3) metabolically healthy overweight (MHOW), (4) metabolically abnormal overweight (MAOW), (5) metabolically healthy obese (MHO), and (6) metabolically abnormal obese (MAO).

Type 2 diabetes was diagnosed based on previous medical records or a blood glucose level ≥126 mg/dL and/or HbA1c ≥6.5% on the blood test [26]. The FLI was calculated according to the algorithm proposed by Bedogni et al. [20]: FLI = (e^0.953 × loge (triglycerides) + 0.139 × BMI + 0.718 × loge (GGT) + 0.053 × waist circumference − 15.745)^ / (1 + e^0.953 × loge (triglycerides) + 0.139 × BMI + 0.718 × loge (GGT) + 0.053 × waist circumference − 15.745^) × 100(1)

All samples were managed and provided by the Regional University Hospital of Málaga -IBIMA (Instituto de Investigación Biomédica de Málaga) Biobank, which pertains to the Andalusian Public Health System Biobank, part of the National Biobank Platform (project PT13/0010/0006). All patients participating in the study gave their informed consent and protocols were approved by the institutional ethics committee (Comité Coordinador de Ética de la Investigación Biomédica de Andalucía. Code: IMAP 06062006).

### Statistical Analysis

The quantitative variables are shown as mean and standard deviation or median and 25th and 75th percentiles in the case of non-normal distribution. The quantitative variables were compared using a one-way analysis of variance (ANOVA) and the qualitative variables using the Chi-square test, the Student’s *t*-test, and the Mantel–Haenszel test. In order to determine factors that were independently associated with pathological FLI, multivariate logistic regression analysis was performed using pathological FLI (FLI ≥ 60) as a dependent variable and controlling for confounding variables such as age, sex, tobacco use, sedentarism, and education level. This association is shown as the odds ratio (OR). A receiver operating characteristic (ROC) curve [27] was applied and the area under the curve (AUC) [28] was analyzed to determine whether FLI was a useful biomarker for characterizing metabolically abnormal phenotypes. All confidence intervals were calculated at 95%. A *p*-value of less than 0.05 was considered significant. The data were analyzed using SPSS version 22.0 (SPSS Inc., Chicago, IL, USA).

## 3. Results

Our final sample of 2233 individuals included 1120 men (50.2%) and 1113 women (49.8%) with a mean age of 43.9 ± 15.6 years. The mean BMI was 27.1 ± 5.1 kg/m^2^ and was significantly higher in men than in women (27.3 ± 4.2 vs. 26.8 ± 5.7; *p* = 0.01). Of the total population, 37.6% had a BMI < 25 kg/m^2^ (30.2% of men and 45.2% of women; *p* < 0.001), 39.1% were overweight (47.6% of men and 30.5% of women; *p* < 0.001) and, 23.3% were obese (22.2% of men and 24.3% of women; *p* = 0.3). The anthropometric, demographic, and analytical characteristics of the overall sample and according to metabolic phenotype are shown in Table 1.

The prevalence of metabolically healthy and metabolically abnormal individuals by BMI category is shown in Figure 2. Among normal-weight individuals, 23.5% were MANW and 76.5% were MHNW. In the overweight group, 49.9% were MAOW and 50.1% were MHOW. In the obese group, 72.1% were MAO and 27.9% were MHO.

No differences were observed on LFT between metabolically healthy and metabolically abnormal phenotypes in any of the BMI groups (Table 1). The mean FLI of the total population was 41.1 ± 28.6 and was significantly higher in men than in women (47.1 ± 26.2 vs. 35.0 ± 29.3; *p* < 0.001). The FLI values according to metabolic phenotype are shown in Figure 3. The FLI was significantly higher (*p* < 0.001) in subjects with metabolically abnormal phenotypes across all BMI categories. The mean FLI values according to BMI categories were 16.4 in subjects with normal weight, 43.1 in subjects with overweight, and 77.5 in subjects with obesity. On the other hand, on the correlation analysis, a positive correlation between BMI and FLI (r = 0.864, *p* < 0.001) was found.

The percentage of the individuals with normal weight and overweight with a normal FLI (< 30) was significantly higher (92.4% and 44.4%, respectively) in the metabolically healthy (MH) subjects compared to metabolically abnormal (MA) subjects (normal weight: 67.2% and overweight: 16.6%, respectively; *p* < 0.001). However, there was no difference between phenotypes in the group with obesity. Finally, the proportion of individuals with a pathological FLI (≥60) was significantly higher in the MAOW and MAO groups compared to the MHOW and MHO groups (*p* < 0.001). In the normal weight category, NAFLD was more prevalent in MANW individuals (4% vs. 0.3%), although this finding was not statistically significant (*p* = 0.06; Figure 4) 

Table 2 shows the results of a multivariate analysis to determine the association between FLI and the total population (Table 2) and metabolically abnormal phenotype (Table 2), adjusted for sex, age, smoking, sedentary lifestyle, and education levels. An analysis of the sociodemographic factors revealed that FLI ≥ 60 was positively associated with male gender, older age, a sedentary lifestyle, and a lower educational level in the total population (Table 2).

These results are corroborated in the analysis of the sociodemographic factors for metabolically abnormal phenotypes, which showed a positive association between FLI ≥ 60 and male gender, older age, a sedentary lifestyle, and a lower educational level (Table 2).

In a multivariable model adjusted for sex, age, and waist circumference, a significant correlation was found between FLI ≥ 60 and metabolically abnormal phenotypes in the overweight and obese BMI categories. However, this correlation was not observed in the normal-weight group (Table 3). 

ROC analysis and area under curve (AUC) with its corresponding 95% CI for metabolically abnormal phenotypes with respect to FLI were 0.76 (95% CI: 0.72–0.80) for MANW with a sensitivity of 4% and a specificity of 0.3%, 0.74 (95% CI: 0.71–0.77) for MAOW with a sensitivity of 35% and a specificity of 10%, and 0.70 (95% CI: 0.68–0.73) for MAO with a sensitivity of 89% and specificity of 72%. (Figure 5a–c)**.**

## 4. Discussion

NAFLD is a frequent complication of excess body fat. Although there is a well-established correlation between NAFLD and BMI and MetS, metabolically discordant phenotypes have been described, with thin subjects who present with MetS and NAFLD and subjects with overweight or obesity who present with no metabolic complications or NAFLD. 

NAFLD screening tools that are applicable in community clinical practice without the limitations of radiological methods or the risks of invasive tests (hepatic biopsy) are necessary. Currently, the most widely used NAFLD screening method in primary care is liver biochemistry determinations, but this has limitations. FLI has been proposed as another screening method and has a sensitivity of 80.3% and specificity of 87.3% [29,30].

We found no significant differences on LFT between metabolically healthy and metabolically abnormal individuals in any of the BMI categories. In other words, although LFT are the principal screening method for NAFLD [1], our data indicate that they do not discriminate between metabolic phenotypes in any of the BMI groups. However, it should be noted that 80% of individuals with NAFLD have normal ALT levels [14]. This absence of differences could be explained through two mechanisms. On the one hand, in patients with abnormal metabolism, insulin resistance could account for liver disease severity and LFT levels [31]. On the other hand, in patients with healthy metabolism, a greater influence of genetic factors could be responsible for increased liver disease severity [32] and LFT values [33].

In contrast, FLI was significantly lower in the metabolically healthy individuals, regardless of BMI category. Likewise, the percentage of individuals with NAFLD (defined as FLI ≥ 60) was lower in the metabolically healthy overweight and obese groups. On the multivariate analysis, there was a statistically significant correlation between NAFLD and the MAOW and MAO phenotypes. Nevertheless, in the normal-weight subjects, the findings were not statistically significant, perhaps due to the sample size. 

Currently, it is understood that fat deposits in the liver play a pivotal role in the development of metabolic complications and the state of insulin resistance associated with obesity [34]. In this study, we have observed an association between the presence of NAFLD and the metabolically abnormal phenotypes MAOW and MAO. In addition, ROC curve analyses showed that FLI yields a diagnostic accuracy ≥0.7 for metabolically abnormal phenotypes. The AUC for FLI in MAO patients has greater sensitivity and specificity than in the MAOW or MANW groups. On the other hand, increased liver enzymes concentrations and triglyceride levels have been proposed as metabolic factors that could explain the differences between metabolically healthy and abnormal phenotype [35]. Previous studies have indicated that increased liver fat accumulation is associated with elevated liver enzymes [36]. Thus, our data are in accordance with the notion that liver fat accumulation is a key determinant of metabolically abnormal phenotypes.

Few studies have analyzed the differences in LFT and liver fat quantity (estimated by diverse methods) between different metabolic phenotypes. In general, MHO subjects are characterized as having less visceral [37], liver [38,39], and subcutaneous fat than MAO subjects [40]. In a study conducted on a group of 314 individuals, Stefan et al. [38] observed that subjects with obesity who were insulin resistant had 54% greater liver fat accumulation (quantified by spectroscopy) than insulin-sensitive subjects. These results were also confirmed by a meta-analysis, which demonstrated an association between NAFLD and insulin resistance and diabetes [41]. Messier et al. [42], who also used FLI, found that MHO subjects were characterized by lower levels of liver steatosis. Nonetheless, the MHO individuals in that study had lower levels on LFT, which was not observed in our study. Pajunen et al. [43] reported lower levels on LFT and less liver fat, estimated by the NAFLD score, in the MHO group compared to the MAO group [14]. Du et al. [44] concluded that MHO individuals have a lower risk of developing NAFLD. It has been described that insulin-sensitive subjects have better NAFLD scores than insulin-resistant subjects across all BMI categories [44].

Lifestyle intervention can be effective for treating NAFLD patients and weight loss can revert liver disease. The Mediterranean diet can reduce liver fat even without weight loss and is the most recommended dietary pattern for NAFLD [45].

The observed association between a low education level and physical inactivity with FLI ≥ 60 suggests two possible routes of intervention that might decrease the prevalence of NAFLD and, as a consequence, metabolically abnormal phenotypes.

### Strengths and Limitations

We conducted a population study that included a representative sample of the adult population of our area. Subjects had a broad age range and there was a remarkably high participation rate. Furthermore, all clinical and anthropometric measurements were meticulously documented by professional healthcare providers (physicians and nurses from the health center). Finally, the ease of use in daily clinical practice of the metabolic phenotype criteria measured in this study is useful from an epidemiological and public health perspective. 

The main limitation of our study is that it was an observational study, thus we were unable to establish causal relationships. As a consequence, all potential factors associated with the metabolic phenotypes should be confirmed in future prospective studies. Additionally, the results of our research may not be extrapolated to other populations as it was conducted in an urban, mainly Caucasian population with a low–middle socioeconomic status. The absence of standardized criteria for the diagnosis of metabolic phenotypes is an important limitation for the comparative analysis of the different studies. Most definitions are based on modified clinical criteria of MetS, but a harmonized definition of metabolic phenotypes does not exist yet [46]. It is well-established that the stricter the definition of the metabolically abnormal phenotype, the lower the sensitivity for detecting it [47]. As a result, in order to achieve greater sensitivity, we decided to define the abnormal metabolic phenotype as subjects who met only two of the four criteria of MetS. Waist circumference was excluded due to the fact that it is closely related to BMI [48]. Unlike other studies [49], we did not include parameters of insulin resistance, markers of systemic inflammation, or hormones among the metabolic risk criteria, which may have limited detection of metabolic abnormalities in our population. We excluded individuals with previous/current alcohol dependence (daily alcohol consumption of >60 g in men and >40 g in women). However, we did not perform individualized alcohol consumption estimations. Consequently, this might have overestimated the number of individuals with NAFLD. Furthermore, we did not analyze the presence of viral hepatitis as a cause of steatosis. Finally, we did not carry out any imaging studies to diagnose NAFLD. Although abdominal ultrasound has a sensitivity and specificity of 84.8% and 93.6%, respectively, for diagnosing moderate–severe NAFLD [50], its usefulness as a population screening tool is limited. Moreover, there is significant interobserver variability and difficulties in its use in obese subjects [51]. In contrast, FLI, a mathematical algorithm based on anthropometric and analytical parameters, although not without limitations, has good external validation using 1H-magnetic resonance spectroscopy as a method for identifying individuals with NAFLD [52].

## 5. Conclusions

In conclusion, based on our results, LFT has a poor correlation with metabolic phenotypes. Conversely, we found a good correlation between FLI, as a marker of NAFLD, and metabolically abnormal phenotypes in all BMI ranges. Although central obesity remains the cornerstone of the metabolic derangement evaluation, our study suggests that FLI may be a useful marker for characterizing metabolically abnormal phenotypes in individuals with overweight and obesity, although its usefulness may be limited in normal-weight subjects.

## Figures and Tables

**Figure 1 ijerph-17-03518-f001:**
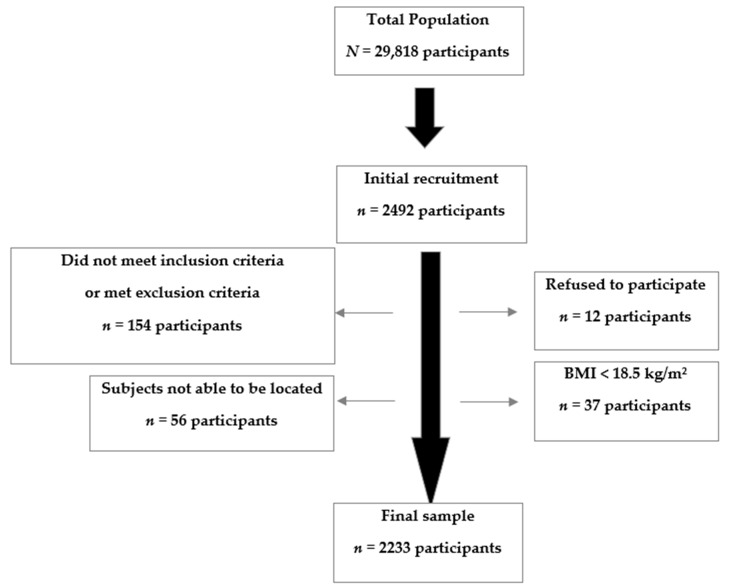
Sample selection process. BMI: Body Mass Index.

**Figure 2 ijerph-17-03518-f002:**
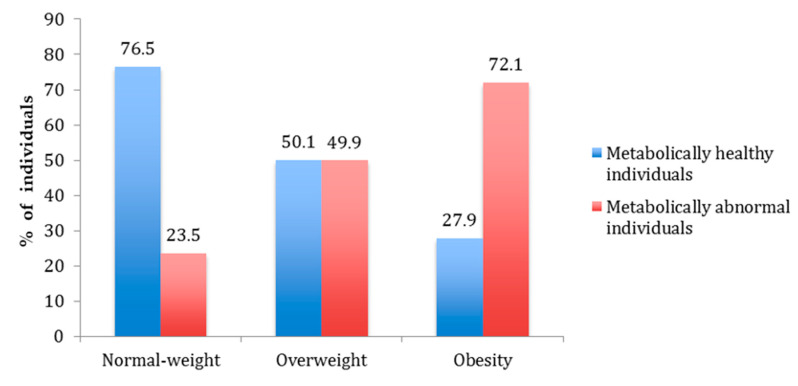
Prevalence of metabolically healthy and metabolically abnormal individuals by body mass index (BMI) category.

**Figure 3 ijerph-17-03518-f003:**
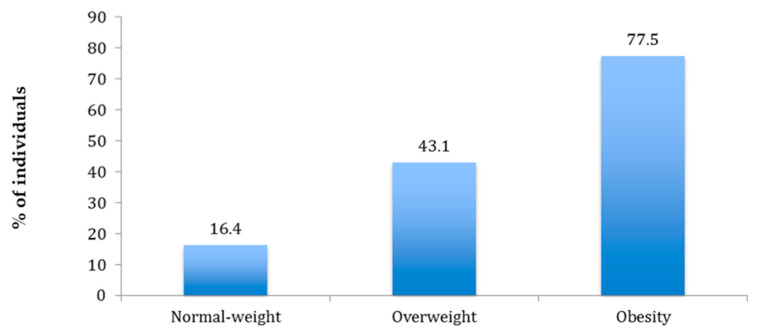
Mean Fatty Liver Index values by body mass index category.

**Figure 4 ijerph-17-03518-f004:**
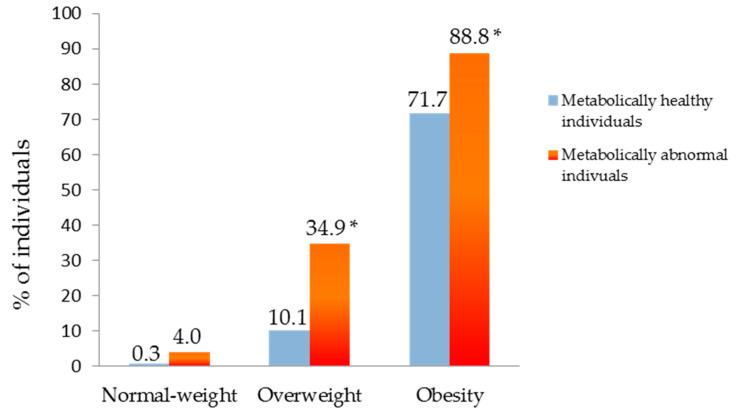
Percentage of individuals with pathologic Fatty Liver Index FLI (≥60) according to metabolic phenotype. * *p* < 0.001.

**Figure 5 ijerph-17-03518-f005:**
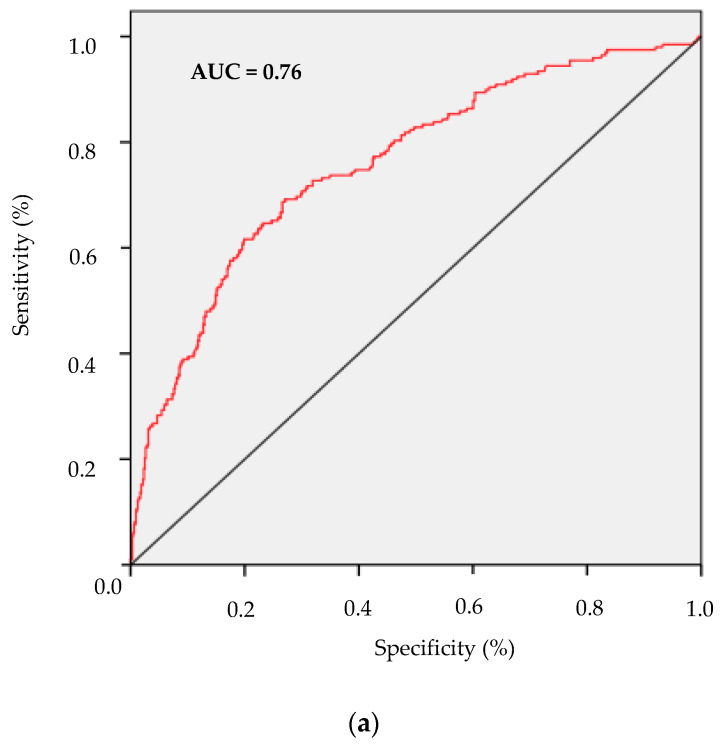
Receiver operating characteristic (ROC) curves. (**a**) metabolically abnormal normal-weight vs metabolically healthy normal-weight, with an area under the ROC curve = 0.76; (**b**) metabolically abnormal overweight vs metabolically healthy overweight, with an area under the ROC curve = 0.74 and (**c**) metabolically abnormal obese vs metabolically healthy obese, with an area under the ROC curve = 0.70.

**Table 1 ijerph-17-03518-t001:** General characteristics (anthropometric, demographic), analytical characteristics, comorbidities, and treatments of the population.

	Total	Metabolically Healthy			Metabolically Abnormal		
		MHNW	MHOW	MHO	MANW	MAOW	MAO
*N* (%)	2233 (100)	643 (28.8)	437 (19.6)	145 (6.5)	198 (8.8)	435 (19.5)	375 (16.8)
General Characteristics							
Age (years)	43.9 ± 15.6	34.7 ± 10.8	40.3 ± 13.6	46.3 ± 15.8	43.6 ± 16.1 *	51.5 ± 15.0 *	54.4 ± 14.3 *
Sex (male/female) (%)	50.2/49.8	34.5/65.5	57.2/42.8	40.0/60.0	59.1/40.9 *	64.8/35.2	50.9/49.1 *
Waist circumference (cm)	90.6 ± 13.5	78.3 ± 7.5	83.4 ± 8.3	102.8 ± 9.8	84.7 ± 8.5 *	94.3 ± 8.8 *	107.1 ± 10.7 *
Weight (kg)	74.0 ± 15.1	61.4 ± 8.4	75.8 ± 10.1	89.4 ± 13.5	64.4 ± 8.5 **	75.4 ± 10.3	91.2 ± 13.2
BMI (kg/m^2^)	27.1 ± 5.1	22.3 ± 1.8	27.1 ± 1.4	33.7 ± 3.9	23.0 ± 1.6 *	27.4 ± 1.4	34.4 ± 4.0 ^†^
Systolic blood pressure (mmHg)	126 ± 16	116 ± 12.9	122 ± 14	127 ± 16	128 ± 13 *	134 ± 15 *	135 ± 15 *
Diastolic blood pressure (mmHg)	75 ± 10	69 ± 9	73 ± 10	76 ± 9	74 ± 9 *	79 ± 10 *	80 ± 10 **
Median-high/low educational level (%)	42.3/57.7	61.1/38.9	46.5/53.5	31.7/68.3	44.4/55.6 *	28.7/71.3 *	23.7/76.3 *
Sedentary lifestyle (%)	76.5	77.8	72.1	82.1	70.7 *	76.6	80.5 *
Smoking (%)	27.6	29.4	24.3	25.5	36.9 *	32.0 ^†^	19.2 *
Analytical parameters							
Glucose (mg/dL)	93.7 ± 26.1	84.3 ± 7.7	85.9 ± 8.6	88.0 ± 7.3	97.1 ± 33.7 *	104.9 ± 36.5 *	106.1 ± 29.3 *
HbA1c (%)	5.7 ± 0.8	5.3 ± 0.3	5.4 ± 0.3	5.4 ± 0.4	6.0 ± 1.1 *	6.0 ±1.1 *	6.1 ± 0.8 *
Creatinine (mg/dL)	0.8 ± 0.2	0.7 ± 0.2	0.8 ± 0.2	0.7 ± 0.2	0.8 ± 0.2 **	0.9 ± 0.3 ^†^	0.8 ± 0.2 *
Uric acid (mg/dL)	4.7 ± 1.4	4.0 ± 1.0	4.7 ± 1.3	4.9 ± 1.2	4.5 ± 1.2 *	5.0 ± 1.5 **	5.5 ± 1.5 *
Total cholesterol (mg/dL)	199.7 ± 40.7	188.6 ± 36.9	194.2 ± 39.1	202.5 ± 39.6	199.8 ± 42.5 **	211.5 ± 42.6 *	210.5 ± 39.4
LDL cholesterol (mg/dL)	124.9 ± 34.5	112.1 ± 31.5	121.0 ± 34.2	125.4 ± 34.3	131.0 ± 34.5 *	137.0 ± 34.0 *	133.7 ± 32.2
HDL cholesterol (mg/dL)	53.3 ± 13.6	59.9 ± 12.5	55.7 ± 11.2	56.7 ± 12.1	47.4 ± 12.9 *	47.9 ± 13.5 *	47.3 ± 12.9 *
Triglycerides (mg/dL)	89 (62–132)	66 (48–92)	76 (57–102)	89 (69–120)	107 (76–160) *	122 (84–178) *	133 (91–187) *
AST (U/L)	23.6 ± 13.0	23.2 ± 13.2	23.6 ± 13.7	25.2 ± 16.1	24.2 ± 12.7	24.1 ± 12.7	22.5 ± 10.6
ALT (U/L)	41.3 ± 17.2	41.3 ± 16.9	40.8 ± 17.4	41.7 ± 15.6	39.7 ± 14.7	42.9 ± 18.0	40.4 ± 18.2
GGT (U/L)	33.6 ± 29.4	34.7 ± 39.4	32.7 ± 22.0	33.3 ± 24.0	30.7 ± 21.2	35.1 ± 26.2	32.8 ± 25.6
Comorbilities an treatments							
Microalbuminuria (%)	6.4	5.6	6.9	4.6	7.7 ^**^	5.0	8.6 *
Prior type 2 diabetes diagnosis (%)	5.6	0	0	0	7.6	10.1	17.6
Type 2 diabetes (%)	9.0	0	0	0	11.1	19.1	25.3
Fatty Liver Index (FLI)	41.1 ± 28.6	13.5 ± 10.0	34.6 ± 16.9	70.4 ± 17.2	25.9 ± 15.7 *	51.5 ± 19.5 *	80.3 ± 15.2 *
Antihypertensive treatment (%)	15.8	1.1	8.0	15.9	15.2 *	23.7 *	41.3 *
Lipid-lowering agents (%)	8.7	1.6	4.1	7.6	8.1	14.0 *	21.1 *

MHNW, metabolically healthy normal-weight; MANW, metabolically abnormal normal-weight; MHOW, metabolically healthy overweight; MAOW, metabolically abnormal overweight; MHO, metabolically healthy obese; MAO, metabolically abnormal obese. * *p* < 0.001; ** *p* < 0.01; ^†^
*p* < 0.05 metabolically abnormal vs. metabolically healthy individuals in body mass index (BMI) categories. HbA1c, glycated hemoglobin A1c; LDL, low-density lipoprotein cholesterol; HDL, high-density lipoprotein cholesterol; AST, Aspartate Aminotransferase; ALT, Alanine Transaminase; GGT, Gamma-glutamyltransferase.

**Table 2 ijerph-17-03518-t002:** Association between sociodemographic factors and fatty liver index (FLI). Total population and Metabolically abnormal phenotypes. Data are showed as odds ratio (OR) and 95% confidence interval (95% CI).

Total Population	FLI < 30	*p*	FLI ≥ 60	*p*
Age	0.94 (0.93–0.94)	<0.001	1.05 (1.04–1.06)	<0.001
Sex (F vs. M)	4.01 (3.13–5.01)	<0.001	0.45 (0.37–0.56)	<0.001
Smoking (Y vs. N)	1.06 (0.85–1.32)	0.58	1.02 (0.81–1.28)	0.86
Sedentary lifestyle (Y vs. N)	0.62 (0.50–0.79)	<0.001	1.45 (1.13–1.86)	0.003
Low education level (Y vs. N)	0.61 (0.49–0.75)	<0.001	1.62 (1.28–2.05)	<0.001
**Metabolically Abnormal Phenotypes**				
Age	0.96 (0.95-0.98)	<0.001	1.03 (1.02-1.04)	<0.001
Sex (F vs. M)	2.55 (1.80–3.60)	<0.001	0.60 (0.45–0.78)	<0.001
Smoking (Y vs. N)	1.19 (0.84–1.70)	0.33	0.81 (0.60–1.08)	0.16
Sedentary lifestyle (Y vs. N)	0.49 (0.34–0.70)	<0.001	1.58 (1.16–2.15)	0.004
Low education level (Y vs. N)	0.64 (0.45–0.93)	0.02	1.50 (1.10–2.05)	0.01

F, Female; M, Male; Y, Yes; N, No.

**Table 3 ijerph-17-03518-t003:** Association between pathological FLI and metabolic phenotypes. Logistic regression model adjusted for sex, age, and waist circumference. Data are shown as odds ratio (OR) and 95% confidence interval (95% CI).

	FLI ≥ 60 (Y/N)	
	OR (95% CI)	*p*
MANW vs MHNW	3.63 (0.65–20.14)	0.14
MAOW vs MHOW	3.07 (1.97–4.76)	<0.001
MAO vs MHO	1.95 (1.08–3.50)	0.03

FLI, fatty liver index; MHNW, metabolically healthy normal-weight; MANW, metabolically abnormal normal-weight; MHOW, metabolically healthy overweight; MAOW, metabolically abnormal overweight; MHO, metabolically healthy obese; MAO, metabolically abnormal obese. Data are showed as odds ratio (OR) and 95% confidence interval (95% CI). OR, odds ratio; CI, confidence interval.

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
