# Peer review of "Utility of Liver Function Tests and Fatty Liver Index to Categorize Metabolic Phenotypes in a Mediterranean Population"

_ijerph, 2020, doi:10.3390/ijerph17103518_

Round 1

Reviewer 1 Report

The authors investigated the utility of liver function test and FLI to discriminate metabolic abnormality.

There are several issues to be resolved before publication.

  1. Waist circumference is important maker to predict the abdominal adiposity. I wonder that why the authors exclude the waist circumference in the metabolic alterations. Please explain in detail the reason for adopting the following criteria.  “1) systolic blood pressure ≥ 130mmHg and/or diastolic blood pressure ≥ 85mmHg (or use of antihypertensive drugs in an individual with a prior history of arterial hypertension) [16], 2) triglycerides ≥ 150 mg/dL (or treatment with a fibrate, nicotinic acid or omega 3 fatty acids) [16], 3) HDL cholesterol < 40mg/dL in men and < 50mg/dL in women (or treatment with a fibrate, nicotinic acid or omega 3 fatty acids) (16), 4) prediabetes defined by blood glucose levels ≥ 100 and ≤ 125 mg/dL and/or HbA1c ≥ 5.7 or ≤ 6.4% [17]. Patients with type 2 diabetes were considered metabolically abnormal, independently of the existence or not of other metabolic alterations." And I wonder that why the authors include the type 2DM.
  2. I could not understand the table 1 and superscripts. And the same superscripts make the authors confused.
  3. It is too difficult to understand the Figure 1. Figure 1a: Prevalence of metabolic phenotypes in the general population; What did you mean? What is different between Figure a and b?. The figure is too redundant.
  4. The authors told that “There was a positive correlation between the BMI and the FLI (r = 0.864, p < 0.001) (Figure 2A).”- Figure 2A is not correlation plot.
  5. Figure 3A and 3B contains similar meaning. The authors should consider how to express the figures in a concise manner.
  6. I also could not understand why the authors choose to show the Table 2. Is it important to show the SES factor in this topic? Please delete that.
  7. Meanwhile, the variables related to SES is too poor. How about income, job, marriage, residual area, or nutritional status?
  8. Table 3: Please re-check the OR and 95% CI values and added the number (%) in MHNW, MANW, MAOW, MHOW, MAO, MHO
  9. Explanation of statistical analysis is too Insufficient

Author Response

We thank you very much for giving us the opportunity to revise our manuscript. We have carefully considered the comments made and agree with most of them. Each comment has been addressed and we have modified the manuscript accordingly. We sincerely hope that the current version of the manuscript will be acceptable for publication in this journal. All changes are shown in red so that they may be easily seen.

On the other hand the authors report that the manuscript, tables and figures have been reviewed by Ms. Claire Conrad, a native English-speaking translator.

Dr. María Rosa Bernal-Lopez

Comments and Suggestions for Authors

The authors investigated the utility of liver function test and FLI to discriminate metabolic abnormality.

There are several issues to be resolved before publication.

  1. Waist circumference is important maker to predict the abdominal adiposity. I wonder that why the authors exclude the waist circumference in the metabolic alterations. Please explain in detail the reason for adopting the following criteria.  “1) systolic blood pressure ≥ 130mmHg and/or diastolic blood pressure ≥ 85mmHg (or use of antihypertensive drugs in an individual with a prior history of arterial hypertension) [16], 2) triglycerides ≥ 150 mg/dL (or treatment with a fibrate, nicotinic acid or omega 3 fatty acids) [16], 3) HDL cholesterol < 40mg/dL in men and < 50mg/dL in women (or treatment with a fibrate, nicotinic acid or omega 3 fatty acids) (16), 4) prediabetes defined by blood glucose levels ≥ 100 and ≤ 125 mg/dL and/or HbA1c ≥ 5.7 or ≤ 6.4% [17]. Patients with type 2 diabetes were considered metabolically abnormal, independently of the existence or not of other metabolic alterations." And I wonder that why the authors include the type 2DM.

Response: In response to the reviewer, the author excluded waist circumference because it is closely related to BMI [Gomez-Huelgas R, Narankiewicz D, Villalobos A, Wärnberg J, Mancera-Romero J, Cuesta AL, Tinahones FJ, Bernal-Lopez MR. Prevalence of metabolically discordant phenotypes in a mediterranean population-The IMAP study. Endocr Pract. 2013;19:758-68]. Most studies define metabolic phenotypes based on the criteria of metabolic syndrome [Pajunen P, Kotronen A, Korpi-Hyövälti E, Keinänen-Kiukaanniemi S, Oksa H, Niskanen L, Saaristo T, Saltevo JT, Sundvall J, Vanhala M, Uusitupa M, Peltonen M. Metabolically healthy and unhealthy obesity phenotypes in the general population: the FIN-D2D Survey. BMC Public Health. 2011;11:754; Koster A, Stenholm S, Alley DE, Kim LJ, Simonsick EM, Kanaya AM, Visser M, Houston DK, Nicklas BJ, Tylavsky FA, Satterfield S, Goodpaster BH, Ferrucci L, Harris TB; Health ABC Study. Body fat distribution and inflammation among obese older adults with and without metabolic syndrome. Obesity (Silver Spring). 2010;18(12):2354-61]. However, in pursuit of clinical coherence, the authors opted to define subjects as metabolically abnormal when they met 2 or more metabolic syndrome criteria (excluding waist circumference and modifying the blood glucose disorders) or if they have diabetes, regardless of whether they met other criteria or not [Kip KE, Marroquin OC, Kelley DE, Johnson BD, Kelsey SF, Shaw LJ, Rogers WJ, Reis SE. Clinical importance of obesity versus the metabolic syndrome in cardiovascular risk in women: a report from the Women's Ischemia Syndrome Evaluation (WISE) study. Circulation. 2004 Feb 17;109(6):706-13]. Some authors have used the abdominal obesity criterion based on waist circumference, instead of the BMI, in order to determine metabolic phenotypes [Velho S, Paccaud F, Waeber G, Vollenweider P, Marques-Vidal P. Metabolically healthy obesity: different prevalences using different criteria. Eur J Clin Nutr. 2010;64:1043-51]. There is a high correlation between BMI and waist circumference, especially in the obese population [Messier V, Karelis AD, Prud'homme D, Primeau V, Brochu M, Rabasa-Lhoret R. Identifying metabolically healthy but obese individuals in sedentary postmenopausal women. Obesity (Silver Spring). 2010;18:911-7], so the waist circumference measurement could be especially important in subjects whose BMI is in the normal or overweight range.

Since there is no consensus on the criteria for the metabolically healthy phenotype, the authors defined the metabolically abnormal phenotype as subjects who have diabetes or ≥2 metabolic abnormalities of those included in the harmonized criteria of the International Diabetes Federation (IDF) for the diagnosis of metabolic syndrome [Alberti KG, Eckel RH, Grundy SM, Zimmet PZ, Cleeman JI, Donato KA, Fruchart JC, James WP, Loria CM, Smith SC Jr; International Diabetes Federation Task Force on Epidemiology and Prevention; Hational Heart, Lung, and Blood Institute; American Heart Association; World Heart Federation; International Atherosclerosis Society; International Association for the Study of Obesity. Harmonizing the metabolic syndrome: a joint interim statement of the International Diabetes Federation Task Force on Epidemiology and Prevention; National Heart, Lung, and Blood Institute; American Heart Association; World Heart Federation; International Atherosclerosis Society; and International Association for the Study of Obesity. Circulation. 2009;120(16):1640-5].

  1. I could not understand the table 1 and superscripts. And the same superscripts make the authors confused.

Response: We agree with the reviewer; the superscripts are confusing. The authors have modified Table 1.

  1. It is too difficult to understand the Figure 1. Figure 1a: Prevalence of metabolic phenotypes in the general population; What did you mean? What is different between Figure a and b?. The figure is too redundant.

Response: The reviewer is correct. Figure 1b has been deleted

  1. The authors told that “There was a positive correlation between the BMI and the FLI (r = 0.864, p < 0.001) (Figure 2A).”- Figure 2A is not correlation plot.

Response: As the reviewer indicates, the text in the manuscript is incorrect. The authors have added a new figure and Figure 2 is now Figure 3. This paragraph has been modified in the “Results” section:

“The FLI values according to metabolic phenotype are shown in Figure 3. The FLI was significantly higher (p<0.001) in subjects with metabolically abnormal phenotypes across all BMI categories. The mean FLI values according to BMI categories were 16.4 in subjects with normal weight, 43.1 in subjects with overweight, and 77.5 in subjects with obesity. On the other hand, on the correlation analysis, a positive correlation between BMI and FLI (r = 0.864, p <0.001) was found.”

  1. Figure 3A and 3B contains similar meaning. The authors should consider how to express the figures in a concise manner.

Response: Figure 4A (previously Figure 3A prior to the addition of another figure) has been deleted. Figure 4 now shows the percentage of subjects with pathologic FLI (60) according to metabolic phenotypes.

In the manuscript, this paragraph had been modified in “Results” section: “The percentage of the individuals with normal weight and overweight with a normal FLI (<30) was significantly higher (92.4% and 44.4%, respectively) in the MH subjects compared to MA subjects (normal weight: 67.2% and overweight: 16.6%, respectively; p<0.001). However, there was no difference between phenotypes in the group with obesity. Finally, the proportion of individuals with a pathological FLI (≥60) was significantly higher in the MAOW and MAO groups compared to the MHOW and MHO groups (p<0.001). In the normal weight category, NAFLD was more prevalent in MANW individuals (4% vs. 0.3%), although this finding was not statistically significant (p=0.06) (Figure 4)”

  1. I also could not understand why the authors choose to show the Table 2. Is it important to show the SES factor in this topic? Please delete that.

Response: For The authors considered it important to note the sociodemographic and lifestyle factors that may influence the development of a pathological FLI.

  1. Meanwhile, the variables related to SES is too poor. How about income, job, marriage, residual area, or nutritional status?

Response: These specific data that the reviewer names were not collected in this study.

  1. Table 3: Please re-check the OR and 95% CI values and added the number (%) in MHNW, MANW, MAOW, MHOW, MAO, MHO

Response: Table 3 shows data on the logistic regression adjusted for sex, age, and waist circumference for the different phenotypes. We believe that perhaps the title of the table is confusing. The authors have modified this Table’s title as follows:

“Table 3. Association between pathological FLI and metabolic phenotypes. Logistic regression model adjusted for sex, age and waist circumference. Data are shown as odds ratio (OR) and 95% confidence interval (95%CI).”

  1. Explanation of statistical analysis is too Insufficient

Response: The statistical analysis section has been modified and additional information has been added.

2.1. Statistical Analysis

The quantitative variables are shown as mean and standard deviation or median and 25th and 75th percentiles in the case of non-normal distribution. The quantitative variables were compared using the one-way analysis of variance (ANOVA) and the qualitative variables using the Chi-square test, the Student’s t-test, and the Mantel-Haenszel test. In order to determine factors that were independently associated with pathological FLI, multivariate logistic regression analysis was performed using pathological FLI (FLI ≥60) as a dependent variable and controlling for confounding variables such as age, sex, tobacco use, sedentarism, and education level. This association is shown as the odds ratio (OR). A receiver operating characteristic (ROC) curve [23] was applied and the area under the curve (AUC) [24] was analyzed to determine whether FLI was a useful biomarker for characterizing metabolically abnormal phenotypes. All confidence intervals were calculated at 95%. A p-value of less than 0.05 was considered significant. The data were analyzed using SPSS, version 22.0. (SPSS Inc., Chicago, IL, USA).           

Reviewer 2 Report

The pourpose of the present study was to analyze the utility of liver function tests and the Fatty Liver Index (FLI), a surrogate marker of nonalcoholic fatty liver disease, in the categorization of metabolic phenotypes in a Mediterranean population. As a results, authors suggests that the FLI may be a useful marker to characterize metabolically abnormal phenotypes in individuals who are overweight or obese. This manuscript is interesting and important, however, this need to revise more several points.

1) Introduction;

Auhors need to be revised more in the Introduction.

Please describe your hypothesis specifically.   2) Methods;   Please add the subjects flow chart using Figure (Methods and/or Results section).   Literature should be used on the significance and judgment of the ROC and AUC numbers in the statistical Analysis.   3) Results; It was not easy to understand in the Table 1. Please revise this with word font.   4) Discussion; Authors need to add more consideration to the determination of ROC and AUC numbers.   5) Limitations; Are there other research limits?  

Author Response

We thank you very much for giving us the opportunity to revise our manuscript. We have carefully considered the comments made and agree with most of them. Each comment has been addressed and we have modified the manuscript accordingly. We sincerely hope that the current version of the manuscript will be acceptable for publication in this journal. All changes are shown in red so that they may be easily seen.

On the other hand the authors report that the manuscript, tables and figures have been reviewed by Ms. Claire Conrad, a native English-speaking translator.

Dr. María Rosa Bernal-Lopez

Reviewer 3 Report

I have read “Utility of liver function tests and fatty liver index to categorize metabolic phenotypes in a Mediterranean population". This study tried to analyze the utility of liver function tests and the Fatty Liver Index (FLI), a surrogate marker of nonalcoholic fatty liver disease. FLI may be a useful marker to characterize metabolically abnormal phenotypes in

individuals who are overweight or obese. This study will provide some useful information, however, there are some concerning points in the scientific quality in this paper.

  1. Authors should introduce the term “liver function” used in this study. In this manuscript, it is difficult for readers what does “liver function” means? Because liver have several functions, what does this term mean is important information for readers.

In this context, the aim of this study “to analyze the utility of liver function tests and the Fatty Liver Index (FLI), a surrogate marker of nonalcoholic fatty liver disease, in the categorization of metabolic phenotypes in a Mediterranean population” should be modified. In addition,

  1. Use correct references of Introduction section. First, authors mentioned “Due to the obesity pandemic, nonalcoholic fatty liver disease (NAFLD) is currently the most common cause of abnormal liver function tests [1]”. However, there is no comment about liver function in this reference paper. Therefore, what does “abnormal liver function test” means is not clear. Second, reference no.2 and 3 do not have adequate information. “NAFLD affects approximately 25% of the global population” showed in your reference no. 4 and, and I could not find data about “up to 80% of obese [3].” In reference no. 3. Please update these references to provide accurate information. Third, author mentioned “over 30.6% became metabolically unhealthy obese after 10 years [10,11].” in page 2 (Please add line no. throughout the manuscript). In ref no 11, the median follow up time was 8.2 years (range 5.5- 10.3), so after 10 years is overstate.
  2. The term fatty liver is not equal NAFLD. Most the most important difference is whether it contains alcoholic fatty liver disease or not. In reference 12 and 13, those authors use fatty liver instead of NAFLD as the term because they did not exclude the subjects with alcohol habits in those study. Although mean alcohol intake was low, the correct term should be used. In addition, in reference no. 14, there is data participant without excessive alcohol consumption. This data is more likely to reflect population of NAFLD than that of total participant that authors chosen. In Introduction section page 2 paragraph 4 line 3, NAFLDA is mistype?

  1. It is hard to understand what is unknown until now in this manuscript. Please emphasize that there is a limited or no-evidence that showed fatty liver index-based categorization of the metabolic phenotypes in Introduction section.

  1. Page 3 line 6, reference (16) should be [16](bold).

  1. Table 1 should be more lager font. Legend of table 1 is not clear. One mark (asterisk) should be mean one thing. Don’t use asterisk for metabolically abnormal vs. metabolically healthy individuals in BMI categories and male/female relationship in metabolically healthy phenotypes. What dose male/female relationship mean?
  2. Figure 1a and Figure 2b are already shown in table 1, please delete it.
  3. Figure 1 to 3, please add unit (% ?) in y axis.
  4. Table 2 and 3, What does this value (e.g. Age 0.94 (0.93-0.94) )mean? Please inform that in legend.
  5. Table 2 and 3, multivariate analysis for determined the association between FLI and metabolically abnormal phenotype, adjusted for sex, age, smoking, sedentary lifestyle, and education levels.

Author Response

We thank you very much for giving us the opportunity to revise our manuscript. We have carefully considered the comments made and agree with most of them. Each comment has been addressed and we have modified the manuscript accordingly. We sincerely hope that the current version of the manuscript will be acceptable for publication in this journal. All changes are shown in red so that they may be easily seen.

On the other hand the authors report that the manuscript, tables and figures have been reviewed by Ms. Claire Conrad, a native English-speaking translator.

Dr. María Rosa Bernal-Lopez

Comments and Suggestions for Authors

I have read “Utility of liver function tests and fatty liver index to categorize metabolic phenotypes in a Mediterranean population". This study tried to analyze the utility of liver function tests and the Fatty Liver Index (FLI), a surrogate marker of nonalcoholic fatty liver disease. FLI may be a useful marker to characterize metabolically abnormal phenotypes in individuals who are overweight or obese. This study will provide some useful information, however, there are some concerning points in the scientific quality in this paper.

  1. Authors should introduce the term “liver function” used in this study. In this manuscript, it is difficult for readers what does “liver function” means? Because liver have several functions, what does this term mean is important information for readers.

Response: As per the reviewer’s suggestion, this sentence has been added to “Introduction” section: “The liver is the largest gland in the human body and plays a role in many functions, such as bile production as well as absorbing and metabolizing bilirubin, fats, proteins, and carbohydrates.”

In this context, the aim of this study “to analyze the utility of liver function tests and the Fatty Liver Index (FLI), a surrogate marker of nonalcoholic fatty liver disease, in the categorization of metabolic phenotypes in a Mediterranean population” should be modified. In addition,

  1. Use correct references of Introduction section. First, authors mentioned “Due to the obesity pandemic, nonalcoholic fatty liver disease (NAFLD) is currently the most common cause of abnormal liver function tests [1]”. However, there is no comment about liver function in this reference paper. Therefore, what does “abnormal liver function test” means is not clear. Second, reference no.2 and 3 do not have adequate information. “NAFLD affects approximately 25% of the global population” showed in your reference no. 4 and, and I could not find data about “up to 80% of obese [3].” In reference no. 3. Please update these references to provide accurate information. Third, author mentioned “over 30.6% became metabolically unhealthy obese after 10 years [10,11].” in page 2 (Please add line no. throughout the manuscript). In ref no 11, the median follow up time was 8.2 years (range 5.5- 10.3), so after 10 years is overstate.

Response: The reviewer is right. The references were incorrect. We have corrected them and the data have been clarified.

  1. The term fatty liver is not equal NAFLD. Most the most important difference is whether it contains alcoholic fatty liver disease or not. In reference 12 and 13, those authors use fatty liver instead of NAFLD as the term because they did not exclude the subjects with alcohol habits in those study. Although mean alcohol intake was low, the correct term should be used. In addition, in reference no. 14, there is data participant without excessive alcohol consumption. This data is more likely to reflect population of NAFLD than that of total participant that authors chosen. In Introduction section page 2 paragraph 4 line 3, NAFLDA is mistype?

Response: As per the reviewer’s suggestion, the "Introduction" section has been modified and the mistakes have been corrected. The follow paragraph has been added:

“There is limited data in the medical literature about NAFLD in the different metabolic phenotypes. In a recent publication, excess adiposity, even without accompanying metabolic health status, may contribute to fibrosis progression in NAFLD [19]. Among all the methods to diagnose NAFLD, we emphasized FLI because it is a non-invasive test that easy to perform and is highly accessible in clinical practice. On the other hand, not all individuals with the same BMI have the same metabolic profile and several studies have attempted to establish criteria characterizing each one. With this background, the aim of this study was to analyze the utility of liver biochemistry and FLI in the categorization of the metabolic phenotypes in a Mediterranean population.”

  1. It is hard to understand what is unknown until now in this manuscript. Please emphasize that there is a limited or no-evidence that showed fatty liver index-based categorization of the metabolic phenotypes in Introduction section.

Response: As the reviewer suggests, these clarifications have been added to the “Introduction” section (see point 3)

  1. Page 3 line 6, reference (16) should be [16](bold). 

Response: This mistake has been corrected.

  1. Table 1 should be more lager font. Legend of table 1 is not clear. One mark (asterisk) should be mean one thing. Don’t use asterisk for metabolically abnormal vs. metabolically healthy individuals in BMI categories and male/female relationship in metabolically healthy phenotypes. What dose male/female relationship mean?

Response: We agree with the reviewer. The authors have modified Table 1.

  1. Figure 1a and Figure 2b are already shown in table 1, please delete it.

Response: As the reviewer suggests, these figures have been deleted.

  1. Figure 1 to 3, please add unit (% ?) in y axis.

Response: The units for the y-axis have been added in Figures 1-3.

  1. Table 2 and 3, What does this value (e.g. Age 0.94 (0.93-0.94)) mean? Please inform that in legend.

Response: Clarification on the values have been added to the tables’ legends (Table 2 and Table 3)

  1. Table 2 and 3, multivariate analysis for determined the association between FLI and metabolically abnormal phenotype, adjusted for sex, age, smoking, sedentary lifestyle, and education levels.

Response: As the reviewer states, a multivariate analysis was used for determining the association between FLI and metabolically abnormal phenotypes, adjusted for sex, age, smoking, sedentary lifestyle, and education levels has been performed.

FLI <30

p

FLI ≥60

p

Age

0.96 (0.95-0.98)

<0.001

1.03 (1.02-1.04)

<0.001

Sex (F vs. M)

2.55 (1.80-3.60)

<0.001

0.60 (0.45-0.78)

<0.001

Smoking (Y vs. N)

1.19 (0.84-1.70)

0.33

0.81 (0.60-1.08)

0.16

Sedentary lifestyle (Y vs. N)

0.49 (0.34-0.70)

<0.001

1.58 (1.16-2.15)

0.004

Low education level (Y vs. N)

0.64 (0.45-0.93)

0.02

1.50 (1.10-2.05)

0.01

Reviewer 4 Report

The Manuscript entitled " Utility of liver function tests and fatty liver index to categorize metabolic phenotypes in a Mediterranean population" analyses the utility of some surrogate markers of NAFLD, examining a sample of adult population referred to a health centre. 

In general, the article is interesting, however I have some concerns:

  • Authors should better describe the enrolment of patients. Was the process  consecutive? 
  • Exclusion criteria included severe diseases. Authors should specify how were the mild diseases accepted.
  • Please discuss the role of   Mediterranean diet.  It is surprising that authors did not comment on  its nutritional role in the discussion section. 

Author Response

Manuscript ID: ijerph-787347

Title: Utility of liver function tests and fatty liver index to categorize metabolic phenotypes in a Mediterranean population

Authors: Dariusz Narankiewicz, Josefina Ruiz-Nava, Veronica Buonaiuto, M Isabel Ruiz-Moreno, M Dolores Lopez-Carmona, Luis Miguel Perez-Belmonte, Ricardo Gomez-Huelgas*, M Rosa Bernal-Lopez*

We thank you very much for giving us the opportunity to revise our manuscript. We have carefully considered the comments made and agree with most of them. Each comment has been addressed and we have modified the manuscript accordingly. We sincerely hope that the current version of the manuscript will be acceptable for publication in this journal. All changes are shown in red so that they may be easily seen.

On the other hand the authors report that the manuscript, tables and figures have been reviewed by Ms. Claire Conrad, a native English-speaking translator.

Dr. María Rosa Bernal-Lopez

Round 2

Reviewer 1 Report

The authors correct the manuscript accordingly.

Author Response

Journal IJERPH (ISSN 1660-4601)

Manuscript ID ijerph-787347

Type Article

Title Utility of liver function tests and fatty liver index to categorize metabolic phenotypes in a Mediterranean population

AuthorsDariusz Narankiewicz, Josefina Ruiz-Nava, Veronica Buonaiuto, M Isabel Ruiz-Moreno, M Dolores Lopez-Carmona, Luis Miguel Perez-Belmonte, Ricardo Gomez-Huelgas*, M Rosa Bernal-Lopez*

We thank you very much for giving us the opportunity to revise our manuscript. We have carefully considered the comments made. We sincerely hope that the current version of the manuscript will be acceptable for publication in this journal.

Reviewer 2 Report

This manuscript will be published as soon as possible.

Because this article has revised based on the reviewers comments.

Author Response

Manuscript ID: ijerph-787347

Title: Utility of liver function tests and fatty liver index to categorize metabolic phenotypes in a Mediterranean population

Authors: Dariusz Narankiewicz, Josefina Ruiz-Nava, Veronica Buonaiuto, M Isabel Ruiz-Moreno, M Dolores Lopez-Carmona, Luis Miguel Perez-Belmonte, Ricardo Gomez-Huelgas*, M Rosa Bernal-Lopez*

We thank you very much for giving us the opportunity to revise our manuscript. We have carefully considered the comments made. We sincerely hope that the current version of the manuscript will be acceptable for publication in this journal.

Dr. María Rosa Bernal-Lopez

Reviewer 3 Report

I have read revised version of manuscript.

  1. Authors should mention the term “liver function” used in this study. I mean what data reflect liver function in this study. I think authors used GOT, GPT, and GGT levels (liver biochemistry?) as liver function. Do my understanding is correct? Authors add the sentence in the introduction section, as follows” The liver is the largest gland in the human body and plays a role in many functions, such as bile production as well as absorbing and metabolizing bilirubin, fats, proteins, and carbohydrates”. However, which data shown in the study reflect liver function is not clear. Therefore, I strongly recommend author define the data that authors mentioned as liver function in this study. In addition, authors should add brief introduction why authors want to compare these data and FLI for predicting NAFLD. I realized that authors already added some sentence in abstract and Introduction section as follows: “Liver biochemistry is poorly correlated with metabolic phenotypes. Conversely, a good correlation between FLI, as a marker of NAFLD, and metabolically abnormal phenotypes in all BMI ranges was found.” and “With this background, the aim of this study was to analyze the utility of liver biochemistry and FLI in the categorization of the metabolic phenotypes in a Mediterranean population.” In those sentences, authors already mentioned “liver biochemistry”, so I recommend to add the recent understanding of relationship between liver biochemistry and diagnosis method of NAFLD.

  1. Page 3 line 16, reference [21] should be [21](bold).

Page 3 line 25, reference(16) should be [16](bold).

Please add line no. in this manuscript. It will become the manuscript easy to read.

  1. Figure 1 to 3, please add unit (% ?) in y axis. I could not find them.

Author Response

Manuscript ID: ijerph-787347

Title: Utility of liver function tests and fatty liver index to categorize metabolic phenotypes in a Mediterranean population

Authors: Dariusz Narankiewicz, Josefina Ruiz-Nava, Veronica Buonaiuto, M Isabel Ruiz-Moreno, M Dolores Lopez-Carmona, Luis Miguel Perez-Belmonte, Ricardo Gomez-Huelgas*, M Rosa Bernal-Lopez*

We thank you very much for giving us the opportunity to revise our manuscript. We have carefully considered the comments made by the reviewer and agree with most of them. Each comment has been addressed and we have modified the manuscript accordingly. We sincerely hope that the current version of the manuscript will be acceptable for publication in your journal. All changes are shown in blue so that they may be easily seen.

On the other hand the authors report that the manuscript, tables and figures have been reviewed by Ms. Claire Conrad, a native English-speaking translator.

Dr. María Rosa Bernal-Lopez
